# Application of Wearable Sensors in Actuation and Control of Powered Ankle Exoskeletons: A Comprehensive Review

**DOI:** 10.3390/s22062244

**Published:** 2022-03-14

**Authors:** Azadeh Kian, Giwantha Widanapathirana, Anna M. Joseph, Daniel T. H. Lai, Rezaul Begg

**Affiliations:** 1Institute for Health and Sport, Victoria University, Melbourne, VIC 3000, Australia; giwantha.widanapathirana@live.vu.edu.au (G.W.); anna.joseph1@live.vu.edu.au (A.M.J.); daniel.lai@vu.edu.au (D.T.H.L.); rezaul.begg@vu.edu.au (R.B.); 2College of Engineering and Science, Victoria University, Melbourne, VIC 3000, Australia

**Keywords:** powered, ankle exoskeleton, orthosis, robot, wearable, human–machine, sensor, actuation, control

## Abstract

Powered ankle exoskeletons (PAEs) are robotic devices developed for gait assistance, rehabilitation, and augmentation. To fulfil their purposes, PAEs vastly rely heavily on their sensor systems. Human–machine interface sensors collect the biomechanical signals from the human user to inform the higher level of the control hierarchy about the user’s locomotion intention and requirement, whereas machine–machine interface sensors monitor the output of the actuation unit to ensure precise tracking of the high-level control commands via the low-level control scheme. The current article aims to provide a comprehensive review of how wearable sensor technology has contributed to the actuation and control of the PAEs developed over the past two decades. The control schemes and actuation principles employed in the reviewed PAEs, as well as their interaction with the integrated sensor systems, are investigated in this review. Further, the role of wearable sensors in overcoming the main challenges in developing fully autonomous portable PAEs is discussed. Finally, a brief discussion on how the recent technology advancements in wearable sensors, including environment—machine interface sensors, could promote the future generation of fully autonomous portable PAEs is provided.

## 1. Introduction

Powered ankle exoskeletons (PAEs) are robotic devices developed for gait rehabilitation, locomotion assistance, and strength augmentation purposes [1]. Traditionally, when developed for assisting with pathological conditions, PAEs may also be referred to as active ankle–foot orthoses (AAFOs) or powered ankle–foot orthoses (PAFOs) [2]. The PAEs developed for rehabilitation purposes are usually wearable robots utilized in rehabilitation facilities that enable repeated walking training rounds on a treadmill or over ground to improve the recovery of the lower-limb motor function in patients suffering from neurological disorders such as stroke, cerebral palsy, and spinal cord injuries. Assistive PAEs, on the other hand, aim to help people with gait disorders affecting the ankle joint caused by ageing, trauma, or neurological conditions to overcome their movement limitations and retrieve a normal and safe gait pattern during their locomotion in daily life. As reported by the World Health Organization [3], about 15% of the total population across the globe experience some form of disability such as muscle weakness, partial or full paralysis or mobility limitation in the lower limb. Therefore, a majority of the currently available PAEs have been developed to address the increasing demand for ankle rehabilitation and assistive devices. However, the application of PAEs is not limited to gait rehabilitation and assistance. Strength augmentation ankle exoskeletons have been developed for powering the ankle joint in healthy users to enhance their performance and reduce the risk of injuries during normal walking, running, or manual handling activities [1,4,5].

The concept of robotic exoskeletons as we know them today goes back to the 1950s when Zaroodny of the U.S. Army Exterior Ballistic Research Laboratory initiated a project on a ‘powered orthopedic supplement’, publishing a report in 1963 [6]. This exoskeleton device was intended to augment the load-carrying abilities of an able-bodied wearer such as a soldier. In the late 1960s, General Electric Research (Schenectady, NY, USA) in collaboration with Cornell University constructed a full-body (680 kg, 30 DoFs) powered exoskeleton prototype funded by the U.S. Office of Naval Research [7]. However, the first powered exoskeleton explicitly developed for the ankle joint might be the early active ankle orthosis presented in 1981 by Jaukovic at the University of Titograd in the former Yugoslavia [8]. This orthosis was actuated using a DC motor placed in front of the wearer’s shin that assisted in dorsi/plantar flexion of the ankle. The footswitches in the soles provided the data required for controlling the device [9]. Long after Jaukovic, in the early 2000s, significant efforts aimed at developing PAEs were initiated by Blaya and Herr at MIT [10], Ferris et al. at the University of Michigan [11], Hollander et al. the Arizona State University [12], and Agrawal et al. at the University of Delaware [13]. Since then, numerous studies have been conducted by many researchers around the globe aimed at developing fully autonomous PAEs.

Regardless of their augmentation or assistive purposes, PAEs must comply with the biomechanics of the anatomical ankle joint and its performance during a gait cycle [14]. According to definitions [15,16], the gait cycle starts with the heel strike of one foot and ends at the next heel strike of the same foot. The gait cycle is usually divided into two main phases: the stance phase and the swing phase (Figure 1). The stance phase begins when the heel of one foot strikes the ground and terminates when the same foot leaves the ground (also known as toe-off). The swing phase is defined as the part of the cycle when the foot is off the ground. The stance phase can be divided into three sub-phases: controlled plantar flexion (heel strike to foot flat), controlled dorsiflexion (foot flat to maximum dorsiflexion), and powered plantar flexion (maximum dorsiflexion to toe-off). During the swing phase, the ankle’s angular position is controlled until it reaches an angle suitable for heel strike. The tibialis anterior, as the major dorsiflexor muscle, is active throughout the swing phase and the loading response. During the loading response and initial swing, the tibialis anterior functions to control the plantar flexion, whereas during the late swing phase it works to maintain the ankle dorsiflexion. As the major ankle plantar flexors, the triceps surae are active during late mid-stance and terminal stance to control dorsiflexion and to generate the force required for the heel to elevate against gravity [17]. The shape and duration of the gait cycle, the kinematic and kinetic characteristics of the ankle movement, and the muscular activity throughout the cycle, may vary between individuals based on their weight, morphology, and health status. An individual’s gait characteristics can also alter from one step to another, depending on the walking pace, locomotion intention, fatigue level, and terrain conditions [18,19,20,21].

Laboratory gait analysis equipment such as force plates, instrumented treadmills, and motion capture systems are the gold standard settings traditionally used for precisely measuring the gait biomechanics and characteristics [16]. However, the growing demand for the light, portable, and wireless measurement tools necessary for conducting field evaluations, as well as the development of portable smart devices, has led to the development of wearable untethered sensors and measurement tools [22,23]. Nowadays, technology innovations in sensor hardware fabrication along with advancements in signal processing and sensor fusion techniques have improved the measurement techniques for bio-signals that describe an individual’s gait biomechanics [24,25,26,27,28,29]. Like many other intelligent wearable devices, PAEs have significantly benefitted from such advancements. It is noteworthy that what makes the PAEs remarkably superior to passive ankle exoskeletons and AFOs is the controllability of the delivered assistance in such intelligent devices. The control hierarchy of a PAE is, therefore, the most critical and complicated component of the device, as it needs to detect the human user’s instantaneous locomotion requirement and ensure the performance of the device is compliant with the user’s intention, while delivering the desired assistance to the user in an optimal fashion [9]. Fulfilling such a complex purpose is not conceivable without the use of an effective integrated sensor system that gathers the required information from the human user and different parts of the exoskeleton in real time. A PAE relies heavily on its sensor system to not only communicate with the human user but also to continuously monitor its performance.

To date, several excellent review papers have been published on lower-limb orthoses and exoskeletons, discussing the design, actuation, and control principles of these devices [14,30,31]. Alqahtani et al., 2021 [1] provided a discussion on different applications of lower-limb robotic exoskeletons. Kalita et al., 2020 [32] systematically reviewed lower-limb robotic-based orthoses and exoskeletons with a section briefly discussing a selected number of PAEs. While Kubasad et al., 2021 [2] has reviewed the design of a number of active and passive orthoses developed for treating drop foot, Jiang et al., 2018 [33] and Shi et al., 2019 [34] focused on the application of these devices in the recovery of stroke patients. Alvarez-Perez, et al., 2019 [35] delivered a review on a selection of seated and walking robots used for ankle rehabilitation. However, in the mentioned reviews, multi-joint exoskeletons and rehabilitation suits have been generally favored over single-joint exoskeletons. Moltedo et al., 2018 [36] provided an exceptional review of studies that investigated the effect of the assistance delivered by PAFOs on healthy and impaired users during walking trials. However, a comprehensive review of the broad range of available PAEs is still missing. Furthermore, the sensor system as a critical component in PAEs and its interaction with the actuation unit and the control hierarchy of the device have not been properly investigated so far. Hence, the current article aims to provide a comprehensive review of how wearable sensors have contributed to the actuation and control of the PAEs developed over the past two decades. Articles concerning seated rehabilitation ankle robots and multi-joint exoskeletons are not included in this review article. For a more detailed explanation of the search and review process, please see the Appendix A.

A PAE is typically composed of an actuation unit, a control unit, a physical frame, and a sensor system (Figure 2). The sensor system is normally composed of human–machine and machine–machine interface sensors and their corresponding electronics such as analog-to-digital converters, digital signal processing units, and microcontrollers with implemented sensor fusion algorithms. The control unit of a PAE usually contains high-level and low-level control schemes. The high-level control scheme uses the information acquired by the human–machine sensors component to produce the actuation command matching the user’s movement intention and requirement. The actuation unit containing an actuator and a power unit then generates the assistive load as commanded. Based on the PAE type, the actuator may provide assistance with dorsiflexion (e.g., avoiding drop foot), plantar flexion (e.g., to power the ankle propulsion), or both. The low-level control scheme monitors the actuator output through the machine–machine sensors component and ensures the output matches with the value commanded by the high-level control algorithm. The physical frame transfers the generated load to the user’s anatomical ankle, while it may or may not allow for a passive movement in the eversion/inversion degree of freedom. The physical frame is required to be light, comfortable to wear, and yet sturdy and durable. Furthermore, the physical frame of a PAE is normally used for housing the actuation, sensors, and control components.

To comprehensively review the selected articles, the following information was extracted from the grouped papers:General information including the exoskeleton purpose and target population, target limb side (bilateral or unilateral), degree of freedom (DoF), and assistance direction (dorsiflexion or plantar flexion or both), portability, and the total weight.Actuation principle and actuator type.Control hierarchy including high-level and low-level control schemes.Sensor system including human–machine and machine–machine sensors

The general information extracted from all 172 reviewed articles is available in Appendix A. A selection of recently developed state-of-the-art PAEs and a summary of their key features are provided in Table 1 for illustration purposes. The selected devices are examples of well-developed PAEs with advanced control algorithms, innovative sensor systems, functional actuation units, and practical wearable physical frames.

## 2. Sensor Technologies Used in Control Hierarchy of the PAEs

The control hierarchy is a core component of a PAE and functions as the decision-making center of the device. To provide optimal assistance or augmentation, the controller of a PAE must cover three important criteria: (I) reliable assessment of the user’s locomotion intention, (II) precise coordination of the timing of assistance with the user’s movement, and (III) generation of an appropriate actuation profile that matches the user’s need and intention while minimizing the unwanted human–machine interaction force wrench. Only if the controller succeeds in meeting these criteria can the robotic device achieve its assistive or augmentation goals [9]. Therefore, designing, developing, and implementing effective control strategies has been the focus of many studies in the field of PAEs [14,44]. The control hierarchy of a PAE can be divided into a high-level control strategy and a low-level control strategy, based on the purposes they fulfil. The higher-level control detects human motion intentions and requirements, and then generates the appropriate displacement or torque command. The low-level control, on the other hand, ensures that the desired command is tracked by the exoskeleton precisely and that the actuator’s output does not cause an interaction force wrench [45].

The decision-making process of a wearable robot (Figure 3) such as a PAE begins with the signal acquisition by the sensors. If acquired in analog format, the signal is first converted to digital format using an analog-to-digital (A/D) convertor. Then, the digital signals acquired from all sources are processed and combined throughout the sensor fusion procedure. Sensor fusion is the combining of sensory data or data derived from sensory data such that the resulting information is, in some sense, better than would be possible if these sources were used individually [46]. When multiple signals are obtained from the same type of sensors (e.g., several EMG biosensors), a unimodal sensor fusion algorithm is used, whereas in multimodal systems, multimodal sensor fusion algorithms combine signals from different types of sensors (e.g., combining data collected from EMG, IMU, and FSR sensors) [47]. Sensor fusion can enhance the performance of a sensor system in the following ways:Robustness and reliability: the redundant data generated by multiple sensor units enables the system to provide information in case of partial failure.Extended spatial and temporal coverage: one sensor can look where others cannot and can perform a measurement while others cannot.Increased confidence: a measurement from one sensor is confirmed by measurements from other sensors.Reduced ambiguity and uncertainty: joint information reduces the set of ambiguous interpretations of the measured value.Robustness against interference: by increasing the dimensionality of the measurement (e. g., measuring the desired quantity with optical encoders and IMUs), the system becomes less vulnerable to interference.Improved resolution: when multiple independent measurements of the same property are fused, the resolution of the resulting value is better than for a single sensor measurement [46].

**Figure 3 sensors-22-02244-f003:**
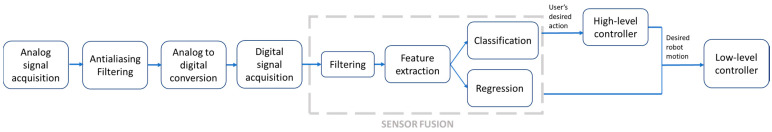
Generic illustration of signal processing and decision-making process in a wearable robot. Adapted with permission from ref. [47]. Copyright 2015 Elsevier.

Filtering is the first preprocessing stage of sensor fusion and includes removing all components of the raw digital signal except those in a defined passband (e.g., 20–500 Hz for EMG). This removes low-frequency mechanical artifacts and high-frequency aliasing effects from the signal. Moreover, to eliminate noise with frequencies within the defined passband, other techniques such as notch filtering and spatial filtering may also be used [47,48]. In the next step, useful information (features) is extracted from the filtered signals using a broad range of mathematical methods via the feature extraction process. The selected features do not need to have the same sampling frequency as the raw signals. Then, further classification and/or regression processing may be utilized on the extracted features. Classification assigns a discrete label to extracted features (e.g., stance or swing phase) while regression converts features to continuous values (e.g., joint angle). Robot control takes the results of classification and/or regression and converts them into the command given to the wearable robot’s actuators [47]. In this section, different types of high-level and low-level control algorithms employed in the reviewed PAEs, as well as the role of wearable sensors in each control strategy, will be discussed.

### 2.1. High-Level Control and Human–Machine Sensors

Apart from Sawicki et al., 2005 [49] and 2006 [50], who investigated the option of controlling the device using a manual **push-button** held by the user or their therapist, all other reviewed articles had developed a form of automatic control. Implementing control strategies is not possible without the use of sensors. The exoskeleton needs to function as an extension of the human body, to power it, and more importantly, support and harmoniously synchronize with it. Bio-inspired high-level controllers, which play a critical role in this regard, rely heavily on biosensors and physical sensors for precisely tracking human motion intentions, in turn leading to human-in-the-loop (HIL) optimization and assist-as-needed (AAN) strategies [51,52]. The high-level control strategies utilized in PAEs can generally be classified into two main trends based on their sensor systems: phase-based controllers and myoelectric-based controllers. **Phase-based control** schemes rely on physical sensors such as force sensitive resistors (FSRs), encoders, and inertial-based sensors, whereas **myoelectric-based control** algorithms use biosensors, i.e., electromyography (EMG) electrodes for collecting the necessary information from the human user. Basic phase-based and myoelectric-based controls comprised the majority of the control options in early robot ankle exoskeletons, though in more advanced prototypes, the basic control schemes were usually combined with a secondary control strategy to enhance the performance of the system (Table 2). Different research groups have taken different approaches to resolving control problems by exploiting the advantages of either of these two main schemes while compensating for the shortcomings via adding other control strategies to their control algorithm. Such remarkable efforts have led to the creation and implementation of novel control algorithms. The section below provides a discussion of these advancements.

#### 2.1.1. Phase-Based Control and Physical Sensors

Phase-based control algorithms aim to track a desired ankle joint torque or angular movement based on the user’s gait phase and kinematic states, measured by a variety of mechanically intrinsic wearable sensors [10,37,38,39,40,41,43,49,53,54,55,56,57,58,59,60,61,62,63,64,65,66,67,68,69,70,71,72,73,74,75,76,77,78,79,80,81,82,83,84,85,86,87,88,89,90,91,92,93,94,95,96,97,98,99,100,101,102,103,104,105,106,107,108,109,110,111,112,113,114,115,116,117,118,119,120,121,122,123,124,125,126,127,128,129,130,131,132,133,134,135,136,137,138,139,140,141,142,143,144,145,146,147,148,149,150,151,152,153,154,155,156,157,158,159,160,161,162,163,164,165,166,167,168,169,170,171,172,173,174,175,176,177,178,179]. Phase-based control schemes are the most frequently utilized high-level algorithms in the reviewed literature due to the simplicity of the algorithms, though when used in their basic form they are not adaptive to individual needs and variations, or even gait mode alternations in the same individual. Phase-based control algorithms, therefore, have been combined with other bio-inspired control strategies to form hybrid high-level control hierarchies that allow for further adaptation and compliance with the user’s biomechanics and locomotion needs [37,54,80,83,86,117,118,141].

**Impedance-based controllers** are an example of such complementary schemes [74,75,136,161,162,163,164,165]. Lopes et al., 2020 [102] developed an adaptive impedance control algorithm that adapts the human–robot interaction stiffness based on the user’s gait phase and state to allow for an assist-as-needed strategy. Nuckols and Sawicki, 2020 [180] used an impedance-based controller to determine the desired exoskeleton torque. The impedance controller was designed to emulate a physical passive elastic element capable of providing plantar flexion torque (rotational stiffness). The desired torque was calculated based on a predefined rotational stiffness and the real-time ankle joint angle.

**Metabolic-rate-based control** [70,96,132,181,182] is perhaps the only variation of phase-based control schemes that takes a physiological input into account in fine-tuning its adaptive gains. Metabolic-rate-based control is a type of human-in-the-loop optimization that aims to select the optimal adaptive values for the control parameters that lead to a minimum steady-state metabolic energy cost during a particular gait mode [70]. The tuning procedure requires laboratory-based equipment including a treadmill, preferably instrumented [96,132], for an accurate real-time ground reaction force measurement and gait event detection, as well as a respirometer device that continuously measures the oxygen intake using metabolic masks [70,96,132]. Yan et al., 2019 [182] used the soleus muscle activities as a physiological indication of the metabolic rate to be minimized during a human-in-the-loop optimization to control their bilateral PAE system. Similarly, Han et al., 2021 [181] presented a metabolic-rate-based control algorithm with a cost function based on surface electromyography signals from four lower-leg muscles instead of respirometry measurements. To construct the cost function, nine gait conditions were defined, where each condition was a combination of different walking speeds, ground slopes, and load weights. Then, ten different assistance patterns were provided by the PAE to the participant for each gait condition. Although such adaptive control schemes offer many benefits in adjusting the assistance level to the user’s needs, they require time-consuming and exhaustive phases of parameter tuning. The procedure usually includes testing a variety of different combinations and values of the control parameters to eventually find an optimized control parameter configuration for a given user, for a selected locomotion mode and a minimized metabolic rate. The system can then be employed by the user without metabolic rate measurement for the selected locomotion mode. Koller at al., 2017 [132] suggested that an instantaneous cost mapping analysis that allows for an estimate of the metabolic cost landscape without the explicit need for steady-state measurements can enable the objective subject-specific comparison of protocols, regardless of which metabolic analysis is used. They developed a novel method for quantifying the confidence in an estimated metabolic cost landscape, which helped them obtain optimal parameter configurations in 20 min, where the standard-practice protocol required 42 min in their previous work [96].

**Reflex-model-based control** is another bio-inspired high-level controller that has attracted the attention of many researchers for developing user adaptive PAEs [68,183,184,185,186,187,188]. A reflex-model-based controller is an adaptive controller that does not need to be tuned for each gait mode. In its ideal form, the model only needs to be fine-tuned once for each individual user, and therefore it can increase user acceptance of robotic exoskeletons for everyday use in dynamic real-life environments. The control algorithm used in this strategy is adapted from the natural neuromuscular reflex mechanism of the human body. The model implements a Hill-type [208] virtual muscle–tendon unit that mimics the biological muscle–tendon unit with similar contractile properties. With respect to the user’s anatomical ankle movement, the length of the virtual muscle–tendon unit changes through a virtual moment arm and stimulates the virtual muscle–tendon unit. The activation signal is generated by a modelled positive force feedback reflex pathway based on the previous force output of the model. When stimulated, the virtual muscle generates force accordingly based on the force–length and force–velocity relationships. The force generated by the virtual muscle–tendon unit is then transformed to a torque value through the virtual moment arm as if the biological muscle–tendon unit had created the ankle torque. Therefore, using this control scheme, the exoskeleton actuator can spontaneously adapt to any dynamic changes in the walking environment or user’s state [183,184].

To achieve their goal, phase-based control algorithms require highly efficient and accurate sensor systems that can precisely detect the occurrence of gait events and estimate the instantaneous gait state in real time. A broad range of wearable physical sensors, also known as mechanically intrinsic sensors, have been used in developing intelligent ankle exoskeletons to fulfil this necessity (Table 3 and Table 4). Force sensitive resistors (FSR), foot-pressure insoles, and footswitches are the most commonly used physical sensors for detecting gait events in PAEs (Table 3). Their application in ankle robot technology development dates back to the initial PAEs [147,148,149]. Kim et al., 2007 [60] and 2011 [61] detected the gait cycle from foot contact signals recorded by FSR sensors that acted as on/off switches and indicated gait events by measuring the voltage drop. Similarly, Hwang et al., 2006 [59] used four FSR sensors and a rotary potentiometer to detect gait phases. Gurney et al., 2008 [209] integrated a set of FSR sensors into shoe insoles to measure the plantar pressure from the main pressure distribution regions of the foot sole: forefoot, toe, and heel, to identify gait events based on the pressure profile of these regions during different states of a normal gait cycle.

Despite their popularity in PAE technology development, insole FSR sensors, footswitches, and pressure films are not without challenges and concerns, as they are prone to cyclic ground impacts and unwanted readings imposed by the shoe [66]. Furthermore, FSR sensors may detect the occurrence of a gait event, although they are not capable of instantaneous estimation of the foot position and orientation throughout a detected gait phase and assessment of exactly what part of the cycle the user is in at a given moment [212]. These shortcomings of FSR sensors and instrumented foot-pressure insoles have led to the development of a variety of sensor fusion strategies and alternative solutions for precise gait event and phase detection in PAEs. For instance, Wang et al., 2020 [57] used an insole-shaped pressure sensor to sense the plantar pressure from three main pressure distribution regions of the sole: forefoot, toe, and heel, to identify gait events. However, the authors found that gait events could not be estimated reliably enough using the insole pressure sensor only, due to the noise and disturbance caused by the user’s shoe. Therefore, they developed a novel sensor fusion strategy that integrated the data collected using both the insole pressure sensor and an inertial measurement unit (IMU) sensor attached to the shoe, to segment the gait cycles more accurately and efficiently. IMU sensors consist of accelerometers, gyroscopes, and in some models, magnetometers. They are portable, light-weight, low power, and body-mountable sensing devices that provide multidimensional acceleration and angular velocity data [213]. However, to use their remarkable capabilities one must first overcome the integration noise and offset voltage drift issues of IMU sensors [214,215,216].

The idea of using IMU sensors in addition to FSR pressure sensors for gait event detection in PAEs is not new. About a decade ago, Caltran et al., 2011 [152] integrated an IMU-based position estimation algorithm with signals from a shoe sole instrumented with three FSR sensors located at the heel, middle, and front of the foot, to achieve fusion-based gait event recognition. To overcome the noise and drift issues, the authors implemented the fusion of information from different sensors using a robust filter that corrected the position estimation within a given range. In this fusion strategy, the position obtained from the accelerometer was used as a redundant measurement with the purpose of correcting the gyroscope-based estimated position. Moreover, the analysis of the FSRs along with the absolute position allowed the precise identification of the events. In a more recent study, Choi et al., 2018 [65] identified a number of issues with regard to using insole FSR sensors for gait state detection. Firstly, FSR sensors are not durable, and repeated stress caused by walking cycles can damage them. Secondly, the ideal positioning of the FSR sensors in the shoe is subjective to individual gait patterns. Finally, the readings of the sensors can be affected by external disturbances and forces applied to the shoe, including those caused by the exoskeleton actuation. As a result, the authors replaced the FSR sensors with a shank-mounted IMU in their next ankle exoskeleton prototype (Seo et al., 2019 [66]), which provided enough information for an advanced recurrent neural network (RNN) gait phase estimation algorithm to detect gait states continuously while protecting the sensors from physical damage caused by repeated ground impacts.

More complex phase-based control schemes rely on more than just gait event detection. Real-time measurement of the user’s lower-leg kinematics provides very valuable information for intelligent control of the PAEs. Potentiometers [10,43,59,102,114,143,147,148,154,155,166,167,170,171] and encoders [38,40,42,58,62,63,64,68,69,72,73,85,86,100,103,104,106,108,109,110,117,118,138,139,142,149,150,151,159,160,161,162,165,173,178,181,185,186,187,188,202,206,207,210] have been broadly deployed in PAEs to measure the angular position and velocity of the exoskeleton ankle joint. The acquired data then can be used for estimating the anatomical ankle joint kinematics based on the mechanical relationship between the anatomical ankle and the exoskeleton hinge joint. In more recent prototypes, IMU sensors have also been used to measure orientation and angular movement of the user’s foot, ankle, and shank [64,119,127,152,211]. Strain sensors can be another option for measuring joint kinematics. Park et al., 2011 [174] and 2014 [175] presented a soft wearable robotic device that used two custom-built strain sensors for measuring the ankle joint angle. The strain sensors were calibrated based on the shank and foot orientation recorded by two nine-DoF IMU sensors during an initial calibration phase. The IMU sensors were then removed, and the joint kinematics were measured merely by the strain sensors. This strategy lowered the power consumption significantly, since a single strain sensor consumes approximately 0.625 mW of power, while one IMU consumes approximately 36 mW. Lee et al., 2021 [64] estimated the ankle kinematics using a combination of an encoder and two IMU sensors placed on the shank and thigh linkages of the exoskeleton. In the PAE presented by Kwon et al., 2019 [87], IMUs attached to the shanks measured the absolute shank angle while soft strain sensors on the knee and the ankle joints measured relative joint angles. Arnez-Paniagua et al., 2017 and 2018 [117,159] placed the IMU sensors on the shank and foot to calculate the shank angle and the translational acceleration of the wearer.

Physical sensors provide very useful information regarding the mechanics of the gait without the need for direct attachment to the user’s skin, since they can technically be mounted on, or even embedded in, the exoskeleton physical frame to collect the required information [47]. However, the measurements are highly affected by the dynamic human–machine interaction between the user’s musculoskeletal system and the physical structure of the exoskeleton. If this complex interaction is not well understood and modelled, the sensor measurements will feed erroneous input to the control unit, which in turn will cause the device to fail to accomplish its mission [203]. In addition, using mechanically intrinsic measurements for control without direct access to the human nervous system has some inherent defects. Mechanically intrinsic sensors in fact measure the outcomes of a physical motion and not the initiators, so they are prone to mechanical delays. A delay in receiving the information will result in the control system not being synchronized with the user’s intention and can cause the user to fight the exoskeleton instead of being assisted by it. Moreover, the control algorithm may not provide suitable assistance in the case of any intended movement by the user that is not included in the predefined actuation profiles [217].

#### 2.1.2. Myoelectric-Based Control and Biosensors

One approach to overcoming the inadequacies of mechanically intrinsic sensors is by accessing the user’s nervous system through surface electromyography (sEMG) and using the sEMG signal as the input to **myoelectric-based controllers**. In the **proportional-myoelectric-based control** strategy, the output of the actuator is set to be proportional to the myoelectric activity of a predefined user’s muscle (e.g., soleus or tibialis anterior) [11,49,68,89,189,190,191,192,193,194,195,196,197,198,199,200,201]. In this control method, the EMG signal collected from the target muscle is processed to calculate the EMG linear envelopes. In traditional proportional myoelectric control, the control signal is computed by multiplying the linear envelope by a constant mapping gain. However, in **adaptive gain myoelectric-based control** [133,134,135,202,203,204,205], the mapping gain is dynamically updated in real time based on changes in EMG recordings and/or the ankle movement state [202,203] to promote the human-in-the-loop assistance strategy. The myoelectric-based controllers exploit surface sEMG as a very effective method to directly and yet non-invasively connect to the user’s nervous system. They are shown to be better synchronized with the user’s physiology compared to the phase-based controllers, since the user has direct control over the timing and magnitude of the actuation. Myoelectric-based control allows the exoskeleton wearer to actively initiate, modify, and stop the actuation, and therefore may promote the user’s neuroplasticity. This type of control does not require a reference value and therefore can potentially be more robust to environmental changes. This advantage makes myoelectric-based control a suitable option for controlling autonomous wearable devices. Nevertheless, the standard sEMG protocols must be in place when EMG biosensors are used as the main human–machine interface measurement tool. A typical sEMG procedure usually begins with preparing the electrode attachment site on the skin to reduce the skin impedance to less than 5 kΩ, locating the target muscle, and correctly placing the electrode on top of the target muscle belly [218]. The process then continues with acquiring EMG signals at a sample rate of no less than 2000 Hz, and applying several signal amplification and filtering steps to the acquired data to finally achieve a high signal-to-noise ratio (SNR), indicating that the acquired data is suitable for use in controlling an exoskeleton [219]. Recent technology advancements have led to the development of wireless sEMG sensors with integrated amplification and filtering circuits that facilitate their deployment in wearable technologies [220].

Despite the valuable benefits of traditional EMG-based controllers mentioned above, it is important to note that proportional-myoelectric-based and adaptive gain myoelectric-based control algorithms produce control commands directly from electromyography readings, while neglecting the highly nonlinear transformations that occur between neural excitation of a muscle (EMG onsets) and the resultant mechanical joint torque generation. Such algorithms do not take into account the nonlinear behavior of an individual muscle–tendon unit or the complex dynamic co-activity of multiple muscles acting on the ankle joint. Accurate estimates of the anatomical ankle joint torque using EMG signals requires data to be acquired from at least the major, if not all, muscles spanning the joint, and requires explicit modelling of the EMG-to-joint torque transformation. Employing a person-specific EMG-driven neuromusculoskeletal modelling approach that estimates muscle forces from electromyography signals through excitation–activation and activation–contraction dynamics in recently developed control schemes has overcome the shortcomings of the traditional myoelectric-based control algorithms. This type of control is known as **neuromechanical-model-based control** [206,207] or **myoelectric neuromuscular-model-based control** and contains a comprehensive sensor fusion procedure that effectively integrates the data from a large set of wearable physical (angular kinematics) and physiological (EMG) sensors to precisely estimate the person’s specific ankle dorsiflexion/plantar flexion torque. Neuromechanical-model-based control can potentially revolutionize the communication between the human and the machine and eventually meet the ultimate goals of PAEs as intelligent user-adaptive robotic devices. However, it is important to note that developing well-performing subject-specific neuromechanical-model-based control for driving a PAE is not a trivial task. EMG-driven models not only inherit the typical electromyography challenges (e.g., noise, crosstalk, skin and movement artifacts, etc.) but also rely on several pre-measurements, parameter tuning, and calibration stages [221,222,223]. For instance, a standard set of isometric tasks would be required for measuring maximum voluntary contraction (MVC) values of the muscles included in model to be used in the EMG normalization procedure. In addition, the absolute MVC value may vary for the same individual from one day to another, due to alternations in environmental or physiological conditions [224]. Furthermore, obtaining MVC values in patients with pathological conditions can be very challenging and requires the development of condition-specific testing protocols [225]. The neuromechanical modelling procedure also requires the development of a musculoskeletal model of the user’s lower limb which is scaled to the user’s anthropometry using static motion capture data. Then, a set of pre-defined effective calibration tasks must be conducted for person-specific optimization of the activation dynamics coefficients and muscle–tendon unit parameters [226,227], which in turn necessitates obtaining high-quality electromyography, motion analysis, and ground reaction force data during the calibration procedure.

Parameters measured using a variety of human–machine interface sensors used in the reviewed PAEs are shown in Table 3. Different sensor types employed for measuring each parameter are grouped and listed together to demonstrate available sensor technologies for measuring a particular parameter. To investigate the sensor types and their implementation in PAEs for detecting the user’s locomotion intention and requirement more deeply, detailed technical information including sensor specifications, measured parameters, and sensor attachment locations is extracted from the reviewed publications and collated in Table 4.

### 2.2. Low-Level Control and Machine–Machine Sensors

The main purpose of a low-level controller in a PAE is to ensure that the command generated by the high-level controller is precisely tracked by the actuation component. The selection of the low-level control scheme is dependent on the actuation principle and actuation mechanism. Choosing the proper actuator for a PAE can be very challenging due to the specific requirements of these devices. PAEs demand high torques and speeds to be able to provide the essential assistance at the ankle joint, given that the joint torque at the anatomical ankle can reach up to 1.5 N·m/kg during normal walking [228]. The main criteria to consider when choosing a suitable actuator for a PAE include: torque-to-mass ratio, power-to-weight ratio, back-drivability, force bandwidth, efficiency, and force density [229,230]. When development of a portable PAE is desired, portability, total weight, and total size of the actuator and its corresponding power source would take a high priority in selecting the actuator type, whereas a powerful and reliable actuation and power source unit might be a better option for tethered rehabilitation PAEs. A further discussion on the challenges in developing portable PAEs is provided in Section 3. Actuators used in PAEs (Table 5) can be classified based on the employed actuation principle [231].

**Pneumatic actuators** are made of pneumatic cylinders or cylinder-like elements with enclosed pistons that can be powered and driven using external air compressors. Four types of pneumatic actuators were used in the reviewed PAEs: pneumatic artificial muscles [11,38,49,50,62,63,88,89,90,91,92,93,95,96,97,98,99,131,132,133,134,135,146,174,175,176,179,190,191,192,193,194,195,196,197,198,199,200,201,203], pneumatic cylinders [43,166,167,168,169,170,171], soft fabric actuators [156,232], and soft fiber braided bending actuators [211]. Pneumatic actuators are cheap and can provide high specific power. Nonetheless, the nonlinearities associated with the compressibility factor of air make the pneumatic actuators very difficult to model and control. Moreover, pneumatic actuators are not ideal choices for portable devices due to the need for external air compressors which are usually very heavy and bulky [233]. Portable air compressors have been used to actuate a number of PAEs [43,131,146,166,167,168,169,170,171,174,175,176], though they do not last long enough to qualify as an everyday usage option.

**Electric actuators** were the most popular actuators deployed in the reviewed PAEs. They were powered by on-board battery packs [37,41,56,57,61,65,66,67,76,77,78,79,80,81,82,83,84,86,87,94,100,101,102,112,113,115,118,119,120,126,127,130,144,145,156,157,158,159,160,172,173,174,175,177,184,187,189,234,235], DC off-board power supply units [10,39,109,116,121,122,128,136,147,148,149], and AC off-board power supply units [72,181]. Eight different types of electric actuation elements were used in the reviewed articles: brushed DC motors [86,115,121,172], brushless DC motors [37,39,40,42,57,64,65,66,76,77,78,79,80,81,82,83,84,100,101,102,103,104,105,106,107,108,109,110,111,112,113,116,117,118,122,123,124,125,127,129,143,147,148,149,150,151,157,158,159,160,161,162,163,164,165,178,184,185,186,187,210], servo DC motors [74,75,87,119,120,141,144,145,177], servo AC motors [53,58,68,69,70,71,72,73,180,181,182,183,202,204], stepper motors [189], permanent magnetic synchronous motors [85], DC voice coil actuators [154,155], and hybrid drive systems [138,139,140,173]. Hybrid drive systems, in fact, are a combination of both hydraulic and electric actuation mechanisms and are often recognized as **electrohydraulic actuators**. Electrohydraulic actuators eliminate the necessity for dedicated hydraulic pumps by replacing the pumps with an electrical motor. In general, electric actuators can provide high speeds, high torques, high force bandwidths, and fast response times. Conversely, they are heavy and suffer from a rather low torque-to-mass ratio [158].

**Series elastic actuators (SEAs)** are made of electrical motors combined with passive elastic elements that store and release kinetic energy as elastic potential energy. Brushed DC motors [67,234,235], brushless DC motors [10,54,55,59,60,61,130,136,152,153,188,207,236], and DC servo motors [137] have been used as the electrical components in PAEs for constructing a variety of SEAs, and springs [110,172,181,236] and tendon-like [149] elements have been used as the passive actuation components. SEAs are designed to control force precisely with spring end positioning [237]. Using SEAs in PAEs has enabled researchers to reduce motor energy and power requirements [12]. SEAs also provide high force amplitudes and force bandwidths [69]. As another solution, Allen, D. P. et al., 2021 [94] recently suggested using **dielectric elastomer actuators (DEAs)** for actuating PAEs. DEAs are made of elastomer films and stretchable electrodes. Typically, elastomer films are sandwiched between stretchable electrodes to create stretchable capacitors. These capacitors can be charged with a constant voltage to create electrostatic forces. DEAs are capable of mimicking the function of artificial muscles and weigh less than pneumatic and electric actuators. However, DEAs require high voltages in the kilovolt range to create a considerable number of electrostatic forces. Additionally, the precise control of DEAs can be very challenging, as their performance can be hindered by the effects of viscoelasticity, the material behavior that causes stress to increase with strain and strain rate. The viscoelastic relaxation effect slows down the DEA strap’s motion over time, and consequently leads to unrepeatability and uncertainty in the actuation unit function [94,238].

**Table 5 sensors-22-02244-t005:** Different types of actuators used in PAEs. Portability of the utilized actuation units is assessed based on their size, weight, and energy source.

Actuation Principle	Actuator Type	Portability	References
Pneumatic	Artificial Pneumatic Muscles (PAM)	Yes	[131,146,174,175,176]
		No	[11,38,49,50,62,63,88,89,90,91,92,93,95,96,97,98,99,132,133,134,135,142,179,190,191,192,193,194,195,196,197,198,199,200,201,203]
	Pneumatic Cylinders	Yes	[43,166,167,168,169,170,171]
	Exosuit Pneumatic Source (Soft Fabric Actuator)	No	[156,232]
	Soft Fiber Braided Bending Actuator	No	[211]
Electric	Brushed DC Motors	Yes	[115,172]
		No	[86,121]
	Brushless DC Motors	Yes	[37,39,40,42,57,65,66,76,77,78,79,80,81,82,83,84,100,101,102,111,112,113,117,118,122,123,127,157,158,159,160,178,184,185,186,187]
		No	[64,103,104,105,106,107,108,109,110,116,124,125,129,143,147,148,149,150,151,161,162,163,164,165,210]
	Servo DC Motors	Yes	[74,75,87,119,120,144,145,177]
		No	[141]
	Servo AC Motors	No	[53,58,68,69,70,71,72,73,180,181,182,183,202,204]
	Stepper Motor	No	[189]
	Permanent Magnetic Synchronous Motors	No	[85]
	Electromechanical DC Voice Coil Actuator	No	[154,155]
	Electrohydraulic Hybrid Drive System	Yes	[173]
		No	[138,139,140]
	Electric Motors (Type Not Specified)	Yes	[56,114]
		No	[41,126,128]
Series Elastic	Brushed DC Motors	Yes	[234,235]
		No	[67]
	Brushless DC Motors	Yes	[130]
		No	[10,54,55,59,60,61,136,152,153,188,207]
		Not Specified	[236]
	Servo DC Motors	No	[137]
	Electric Motors (Type Not Specified)	Yes	[206]
		No	[205]
Dielectric Elastomer	Polyimide Fibers	Yes	[94]

PAEs are nonlinear systems highly subject to a variety of disturbances and uncertainties from the environment and the human user. The complex dynamics of the system and parameter perturbations make the control problem challenging in PAEs. Although a number of reviewed articles utilized **open-loop feed-forward** algorithms for controlling actuation units [94,158,184], the majority of the considered studies developed a form of feedback controller (Table 6). **Classical proportional–integral–derivative (PID)** feedback controllers [39,53,54,55,56,67,68,76,77,78,79,82,86,101,102,108,109,110,114,119,120,121,136,138,139,140,154,155] were the most frequently used control algorithms in the reviewed PAEs due to the simplicity of real-time implementation [239]. Integral, proportional, and derivative feedback is based on the past (I), present (P) and future (D) state of the system Three types of controllers have been devised based on the integral, proportional, and derivative feedback controllers that are widely used in PAEs: P [100,106,107,113,146,172,178,185,186,187,204], PI [85,100,105,178,185,186,187], and PD [10,57,65,66,69,70,71,72,73,81,137,147,148,149,150,151,152,153,161,162,163,164,165,180,181,202,206,207]. Proportional control cannot fully eliminate the disturbance effects. When used with the integral term (I), the proportional–integral controllers can reject the constant disturbances, though they still function poorly with regard to the occurrence of time-varying disturbances. Using the derivative term may assist with this issue, but it also increases the noise sensitivity [44]. A practical solution is to combine an adaptive control algorithm with the PID controller and develop an adaptive PID control [37,64,80,83,84], so that the proportional, integral, and derivative gains can be adapted to the time-varying disturbances [239].

**Adaptive control** algorithms such as active disturbance rejection [40], model reference adaptive control [159], adaptive proxy-based [160], and extremum-seeking control [189] have also been used in more recent PAEs. An **iterative learning** algorithm is another practical option since it does not require an accurate model in order to function [58,68,69,70,71,72,73,131,181,182,202]. Iterative learning is in fact an unsupervised machine learning algorithm that improves the system performance by learning from previous executions [240] when the system executes the same task multiple times. Therefore, it does not rely on knowing the external disturbances beforehand. This type of control has been shown to perform better than the classical PID controller for repetitive movements, e.g., walking on a treadmill [68]. However, iterative learning might not be an ideal solution when the user interacts with a dynamic environment with a varying gait style and pace. In [68], the torque-tracking performances of nine different hybrid control strategies with various combinations of PID, model-based, adaptive, and iterative learning controls were experimentally compared when combined with phase-based, reflex-model-based, or myoelectric-based high-level controllers. The results showed that the combination of proportional control with damping injection (PD) and iterative learning resulted in the lowest errors for all high-level controllers. Based on this finding, in [69,71,72,73,181,202] the authors combined a proportional–derivative (PD) control with an iterative learning scheme to benefit from both strategies. However, it is important to consider the fact the experimental protocol used in [68] included collecting data from only one participant, during walking on a treadmill at 1.25 m/s for one hundred steady-state steps. Therefore, the results may not hold for walking in a dynamic environment with varying gait modes. In a recently developed PAE, a sliding mode control algorithm was employed in combination with an extended state observer (ESO) to benefit from the high level of robustness this algorithm offers [53]. **Sliding mode control (SMC)** is a nonlinear control algorithm based on the variable structure method. This means the state-feedback control law is not a continuous function of time, rather it can switch between different continuous functions to alter the dynamics of a nonlinear system based on the current state. The state-feedback control law is not a continuous function of time. Instead, it can switch from one continuous structure to another based on the current position in the state space. Hence, it does not need to be precise and will not be sensitive to parameter variations [241].

With regard to pneumatic actuators and artificial muscles, the actuation unit is more difficult to model and control compared to electric motors. According to the reviewed literature, there are three methods available for controlling pneumatic actuators using solenoid valves: on–off switch solenoid valves [146,156,166,167,170,171,173,174,175], pulse width modulation (PWM) of the solenoid valves [38,62,63,141,173,188], and proportional pressure regulators with solenoid valves [11,43,49,50,88,89,90,91,92,93,95,96,97,98,99,132,133,134,135,142,168,169,190,191,192,193,194,195,196,197,198,199,200,201,203]. In the proportional pressure regulation method, the air pressure inside the artificial muscles is controlled based on a signal proportional to the plunger displacement. When the plunger is not depressed, no air pressure is supplied to the artificial muscles, and when fully depressed, the control system sets the artificial muscle pressure at the maximum level. In [175], a combined low-level controller was created with a PWM controller, on/off solenoid valves, and a model-based disturbance rejection controller. In [146,176], proportional position controllers were combined with solenoid valves.

Machine–machine interface sensors in PAEs play a critical role in gathering the required information for the low-level controllers (Table 7 and Table 8). Feedback controllers, regardless of their algorithm, are required to continuously monitor the actuator’s output and apply corrective strategies to ensure that the desired assistance will be transferred to the device, and later delivered to the user. Torque sensors [37,76,77,78,79,80,81,82,83,84,85,104,105,106], strain gauges [68,69,73,202], potentiometers, and load cells have been used to measure the actuator-generated torque, while tension sensors [53,182], force sensors [57], load cells [39,58,69,108,109,110,122,123,124,125,126,127,128,129,180,183,204], and strain gauges [72,113,181] have been used for measuring the tension force in the force transmission parts of the exoskeletons, such as the cables and ropes. Current sensors [39,85,161,162] and encoders [39,103,104,106,108,110,111,112,113,115,116,121,131,141,143,147,148,149,150,151,161,162,165,172,188,206,207,210,235,236] are widely used in a variety of electric motors, whereas pressure sensors [62,63,146,156,158,166,171,173,176,232] play a critical role in controlling pneumatic actuators. Parameters measured using a variety of machine–machine interface sensors used in the reviewed PAEs are shown in Table 7. Different sensor types employed for measuring each parameter are grouped and listed together to demonstrate available sensor technologies for measuring a particular parameter. To investigate the sensor types and their implementation in PAEs for measuring the actuation unit outputs more deeply, detailed technical information including sensor specifications, measured parameters, and sensor attachment locations was extracted from the reviewed publications and collated in Table 8.

## 3. Towards Fully Autonomous Portable PAEs

Tethered PAEs featured in more than 56% of the reviewed articles. However, apart from a small number of articles aimed at developing ankle robots to be utilized explicitly in rehabilitation facilities for gait training purposes in stroke survivors [162,163,164,165] or those with incomplete spinal cord injuries [50], the rest of the tethered ankle exoskeletons were developed for research purposes with an ultimate goal of fabricating a portable autonomous device that can be worn on the ankle joint during daily routines. Such tethered exoskeletons should be considered as strong research tools that have significantly enhanced our knowledge of neuromechanical control of human walking [88], the biomechanics of human–machine interaction between the user and the robotic device [134,142], and device controllability and robustness against gait mode changes [96,132,133,134]. Large off-board actuators with wide torque ranges, along with standard laboratory-based motion analysis equipment [96,132,133,134], have enabled researchers to study different aspects of PAE technology development in a controlled laboratory environment without actuation torque limitations. However, despite their remarkable contribution to prompting rapid prototyping and design iteration, tethered ankle robots cannot be used as wearable self-contained devices during the activities of daily living while still in their current state of development.

In moving towards full portability, there are many aspects to be considered in developing PAEs. Current PAEs still have substantial added mass, limited mechanical power, and tethered or limited energy supplies, while a wearable device is expected to be light, comfortable to wear, and have a power source that lasts through the day [111]. Choosing the right actuation unit and its corresponding power source is one of the major challenges facing the development of portable PAEs, as there are trade-offs between power- and torque-to-weight ratios, back-drivability, force bandwidth, efficiency, and portability [9]. Light and portable actuators are not usually capable of delivering the torques required for supporting the ankle joint during normal gait (approximately up to 1.5 N·m/kg [228]), whereas stronger motors consume a larger amount of power and need bulky and heavy battery packs to function throughout the day [242]. Coupling the actuators with gearboxes to increase the torque-to-weight ratios [158] or elastic components that can slowly store some part of the gait energy and quickly release it during push-off has been shown to help with the power-source problem to some extent [100,115,148], though future advancements in chargeable battery technologies [243] are also expected to cause significant modifications in the next generation of PAEs. Optimizing the physical structure of the exoskeleton also plays a very critical role in decreasing the total weight of the device, lowering power consumption, and of course improving the user’s comfort and acceptance. The ideal exoskeleton must reliably transfer the generated actuation to the anatomical ankle, be durable and sturdy yet light and comfortable to wear, and not physically interfere with the user’s movements. To meet these criteria, researchers have been exploring a variety of options from soft robots to different engineering materials and manufacturing methods [1].

### Role of Wearable Sensors in Developing Autonomous Portable Ankle Exoskeletons

Portable ankle exoskeletons would not be possible without wearable sensors that enable researchers to make the required measurements without being restricted to laboratory-based equipment. Deploying insole pressure sensors and footswitches as well as potentiometers and encoders in early PAEs was one of the initial efforts towards freedom from laboratory-based measurement equipment. Blaya et al., 2004 [10] used an Ultraflex system as a replacement for force platforms in clinical gait laboratories. This system was instrumented using six capacitive force transducers with 25 mm square area and less than 3 mm thickness, placed on the bottom of the foot. Each sensor could detect up to 1000 N with a 2.5 N resolution and a scanning frequency of 125 Hz. Ultraflex provided the input for the adaptive impedance control, while a single footswitch was placed in the heel of a shoe worn with the exoskeleton to detect the heel strike approximately 30 ms earlier than the Ultraflex force sensors. In this prototype, a potentiometer was used for measuring the ankle joint plantar flexion/dorsiflexion angle without the need for a motion capture system. A year later, Ferris et al., 2005 [11] used surface EMG signals from the soleus and tibialis anterior muscles to control artificial pneumatic muscles that actuated a robotic ankle exoskeleton based on a proportional-myoelectric-based control scheme. This study proved the feasibility of developing a lightweight PAE that can be controlled using EMG electrodes only. Myoelectric-based controllers were demonstrated to make the timing and amplitude of actuation more aligned with the user’s intention compared to phase-based controllers, as they enable direct access to the user’s nervous system [36].

The selection of wearable sensors depends strongly on the control strategy and its complexity level. For instance, simple phase-based controllers merely need a gait event detection system and an angular movement measurement unit to gather their required information. Such information can be collected from physical sensors such as force sensors, potentiometers, or IMUs simply attached to the device rather than to the user’s skin [36], which makes the exoskeleton more wearable and user-friendly. Therefore, phase-based control algorithms might be a suitable option for ankle exoskeletons that are specifically made for augmenting the ankle power during walking and running in healthy users. In contrast, metabolic-rate-based control schemes, which rely on laboratory-based respiratory measurements to optimize their control parameters, cannot be considered when developing portable autonomous exoskeletons despite their sophisticated human-in-the-loop (HIL) optimization method [70,96,132]. Instead, other wearable biosensors such as surface EMG electrodes, pulse oximetry units, and/or low-profile ultrasonography probes may be used in future exoskeleton technologies for continuous monitoring of the user’s physiological state and for optimizing the assistance provided by the robotic device [244].

An ideal control strategy for a portable ankle exoskeleton should not need repeated time-consuming laboratory-based calibration and parameter tuning processes and is expected to show robust, adaptive behavior in response to changes in both the state of the user and the environment [183]. Biosensors such as surface EMG electrodes play a critical role in effectively detecting changes in the user’s state and/or intention. In addition to electromyography (EMG), other techniques such as mechanomyography (MMG), which measures the sound of the muscles, sonomyography (SMG), which measures muscle thickness [245,246], and ultrasound imaging [247] have been previously explored for controlling robotic prostheses and may soon be investigated for improving biosensors and movement prediction technologies in PAEs. Another near-future advancement in PAEs is expected to involve employing brain–machine interface (BMI) technologies using electroencephalography techniques (EEG) [248] or neural implants to quickly and accurately detect the user’s locomotion intention and generate the appropriate actuation command. Such significant breakthroughs will substantially change the face of current PAEs [52]. Therefore, new technologies such as artificial intelligence (AI), along with biosignal processing and neural technology, will be extremely important for the development of future exoskeleton research [249]. Thanks to corrective and predictive machine learning algorithms [216,250], IMU sensors seem to be a very promising replacement for a wide range of physical sensors such as FSRs, pressure insoles, footswitches, potentiometers, and encoders, as they are shown to be capable of both detecting the gait phases and measuring the limb orientation and movement. The magnetometer and accelerometer components in IMU sensors might also become useful in the near future for evaluating muscle activities without EMG electrodes [251]. Using a limited number of sensors that can measure several parameters has many benefits including decreasing the total weight, lowering the amount of required signal preparation and signal processing, reducing power consumption, decreasing the noise sensitivity, and finally promoting the wearability and portability of the device.

As mentioned above, the ideal control algorithm for PAE must be adaptive to locomotion environment changes. Although the application of environment–machine interface sensors was not observed in the reviewed articles, we expect that terrain recognition and obstacle detection technologies will soon be integrated into the next generation of PAEs, similar to the currently available technologies in robotic prostheses [52,252]. For instance, Fan et al., 2011 [253] used a laser distance sensor in combination with an IMU to identify the movement state and upcoming terrain. The placement of the IMU sensor was investigated for both shank and waist. The results revealed a 98% accuracy in terrain recognition for the waist placement. A laser in combination with an IMU at the waist was also used by Liu et al., 2016 [254] to recognize the terrain. Terrain detection accuracy was above 98%, and the system succeeded in informing the control system about the upcoming terrain change more than 0.5 s before the time required for the control unit to switch mode and adapt to the new terrain. In this study, it was shown that employing environment–machine sensors for terrain recognition also improved the accuracy and the reliability in detecting the user’s intended locomotion mode. To determine the foot orientation with respect to the ground, Scandaroli et al., 2009 [255] placed four infrared sensors beneath the foot. Their proposed system was able to detect objects located up to 0.3 m away. Krausz et al., 2015 [256] developed a depth-sensing system using a Microsoft Kinect. The system was shown to be able to detect the presence of stairs with an accuracy of 98.8% while also accurately estimating the staircase, the distance to the stairs, the angle of intersection, the number of steps, the stair height, and the stair depth. Diaz et al., 2018 [257] mounted a camera on the shank to collect the visual data required for a terrain recognition system. The system achieved an accuracy of 86% in classifying the terrain into six different categories. The results proved the system to be capable of measuring the inclination angle of the terrain.

Obstacle detection technologies used in vehicles [258] and mobile robots [259] are gaining more attention from researchers for potential applications in assistive technologies [252]. Several techniques can be used for this purpose, including ultrasonic sensors (sonar), laser range scanners (LRS), and computer vision (CV) techniques [252]. Costa et al., 2012 [260] employed a stereo imaging technique for developing an obstacle detection and avoidance module to assist visually impaired people while navigating. The required data were fused from an RFID reader placed on the cane (connected via Bluetooth) and a chest-mounted camera, as well as a GPS and Wi-Fi antennas which were built-in components of the mobile computer unit. Similarly, Vlaminck et al., 2013 [261] developed a system that aimed to assist visually impaired people in indoor environment navigation by detecting walls, doors, and stairs, as well as loose obstacles and bumpy parts of the floor, using a 3D-imaging Kinect sensor. The application of lidar and stereo for obstacle detection in both structured (e.g., indoor, road) and unstructured (e.g., off-road, grassy terrain) environments was explored by Kuthirummal et al., 2011 [262]. The provided system aimed to detect scene regions that were traversable and safe for a robot to go to from its current position. The smart shoe presented in a very recent study by Wu et al., 2021 [263] is capable of detecting obstacles using an ultrasonic sensor and an accelerometer that measures the 3D acceleration of the foot to detect obstacles. The system includes a gait events recognition algorithm that detects the motion state of feet. When the foot is in the stance phase, the obstacle detection algorithm is activated. This smart shoe also has a fall detection system, and in the case of a fall incident it will automatically connect to a mobile phone and call the emergency contacts.

Technology advancements in sensor fabrication [24] will significantly enhance the wearability of sensor systems in future ankle exoskeletons. Recently developed technologies in the field of microfabrication, microelectronics, flexible electronics, and nanomaterials combined with wireless communication, Internet of Things (IoT), and signal processing advancements have led to the development of textile-based [264,265,266] and skin-like (epidermal) [266,267] wearable sensors. Nanotechnology and nanomaterials in particular have facilitated the rapid production of highly sensitive wearable sensors for a wide range of applications [268]. These innovations now make it possible to equip future intelligent ankle exoskeletons with compact, light, low-cost, and customizable multifunctional sensors that can be simply mounted in any desired location, from the user’s skin to fabric and flexible materials used in the exoskeleton structure, in order to collect the target signal from the human body, the device, and the environment [269]. This will significantly improve how the smart exoskeleton communicates with the human user and their locomotion surroundings. The incorporation of temperature, humidity, pressure, and touch sensors [270,271] could considerably improve the comfort of wearing the PAE and the alignment of the device with the user’s movement intention and requirement. The deployment of such technologies in recent smart prostheses [272] indicates their near-future applications in exoskeletons and assistive devices.

## 4. Conclusions

Wearable sensor technology has played in a significant role in the development of PAEs throughout the past two decades. Human–machine sensors in both physical and biosensor forms act as the communication media between the robotic device and the human user. They assess the user’s locomotion phase, body position, and orientation, as well as the muscle activities, to inform the device about the user’s movement intention and power requirement, either to augment the user’s performance or to assist them to overcome their locomotion limitations and regain their normal walking pattern. Machine-machine interface sensors also continuously monitor the output of the actuation unit in terms of pressure, current, and motor position, as well as generated torque and force, and inform the low-level control schemes of any errors and disturbances to ensure the actuation unit is precisely following the high-level control commands. The advancement in control algorithms has considerably affected the deployment and application of human–machine sensors. The early generations of PAEs simply relied on FSR on/off switches to detect a gait event and impose a pre-assumed joint torque or joint angle profile to the user, or they used a constant ratio to relate the EMG activity of a single muscle to the ankle joint torque. Nowadays, high-level control schemes have evolved into sophisticated algorithms that take a broad range of data acquired from several sources by a variety of wearable sensors such as IMUs, force sensors, encoders, potentiometers, and wireless EMG biosensors that come with built-in signal processing circuits, and employ a combination of advanced sensor fusion, human-in-the-loop optimization, and assist-as-needed strategies to eventually develop PAEs that people can wear and use as an extension of their own body.

Despite the significant advancements in the development of PAEs, they have not yet been widely adopted by end users. Currently available technologies are still far from providing a fully portable autonomous device that can be worn on the ankle joint on a daily basis and can intelligently adapt its behavior to continuous changes in the user’s locomotion intention and requirement, as well as changes in the surrounding environment and objects. Such advancements require further developments in obtaining information from both the user and the locomotion environment. Rapid technology advancements in biosignal processing are expected to provide new opportunities for exploiting EMG and EEG signals for involving brain–machine interfaces in exoskeleton control systems and improving understanding of the user’s intention and neuromuscular function. This will also significantly improve our current knowledge of how the user’s neuromuscular system alters when interacting with an active intelligent wearable device. Environment–machine sensors are needed for inclusion in future generations of ankle exoskeletons to inform the control hierarchy of terrain changes, obstacles, and tripping hazards. Advancements in machine learning and artificial intelligence will soon modify the way we use the data acquired from physical sensors, enabling extraction of more features with higher accuracy. Finally, technology advancements in sensor fabrication can potentially enhance the wearability of PAEs by providing lightweight, inexpensive, and multifunctional textile-based and skin-like sensor systems, and it is suggested that these should be considered in developing future generations of PAEs.

## Figures and Tables

**Figure 1 sensors-22-02244-f001:**
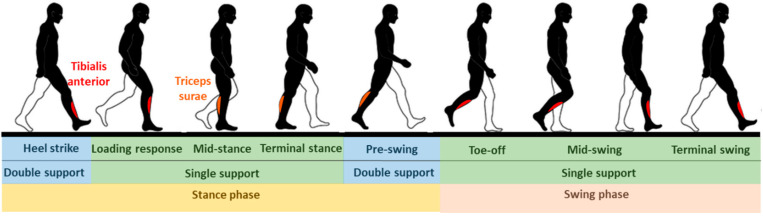
Gait cycle divisions and the activation of the ankle dorsiflexor (tibialis anterior shown in red color) and plantar flexor (triceps surae shown in orange color) muscles during each phase of the gait. Adapted with permission from ref. [16]. Copyright 2006 Elsevier.

**Figure 2 sensors-22-02244-f002:**
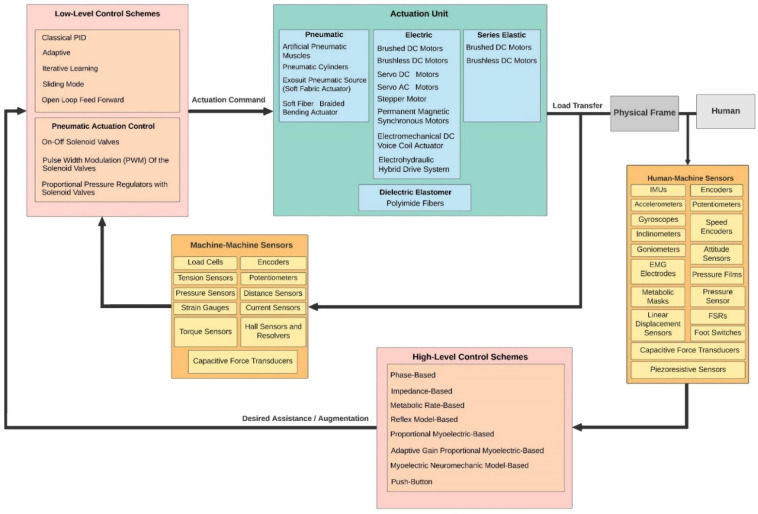
Schematic block diagram of the interaction between the main components of the PAEs. The list of different technologies and methods used in the reviewed literature for developing each component is provided in the corresponding block.

**Table 1 sensors-22-02244-t001:** An example set of state-of-the-art PAEs selected from the reviewed articles that demonstrate advanced control algorithms, innovative sensor systems, functional actuation units, and practical wearable physical frames.

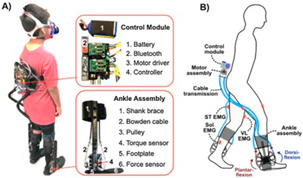 Fang et al., 2020 [37] (copyrights authorized by Elsevier)	Purpose: Assistive Device: Cerebral Palsy, Neuromuscular Impaired, and Parkinson Patients Bilateral: Yes DoF: 1 DoF Plantar Flexion Portability: Portable Weight: 1.85 kg–2.20 kg	High-Level Control Scheme: Phase-based Human-Machine Sensors: FSRs Low-Level Control Scheme: Adaptive PID Machine–Machine Sensors: Torque Sensors Actuation Mechanism: Brushless DC Motors
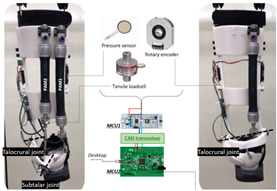 Choi et al., 2020 [38]	Purpose: Assistive Device: Elderly Bilateral: No DoF: 2 DoF Plantar Flexion and Eversion/Inversion Portability: Tethered Weight: 2.14 kg	High-Level Control Scheme: Phase-based Human–Machine Sensors: FSR, Encoder Low-Level Control Scheme: Pulse Width Modulation (PWM) with Solenoid Valves Machine–Machine Sensors: Load Cell Actuation Mechanism: Pneumatic Muscle
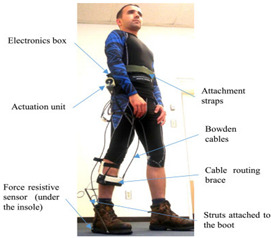 Bougrinat et al., 2019 [39] (copyrights authorized by Elsevier)	Purpose: General Augmentation Bilateral: No DoF: 1 DoF Plantar Flexion Portability: Portable Weight: 2.045 kg	High-Level Control Scheme: Phase-based Human–Machine Sensors: FSR, Low-Level Control Scheme: PID Machine–Machine Sensors: Load Cell, Current Sensor, Encoder Actuation Mechanism: Brushless DC Motors
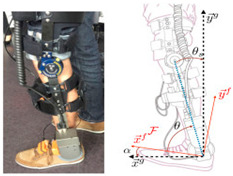 Guerro-Castellanos et al., 2018 [40] (copyrights authorized by Elsevier)	Purpose: Assistive Device: Drop Foot and Paretic Patients Bilateral: No DoF: 1 DoF Dorsiflexion/Plantar Flexion Portability: Portable Weight: 3.5	High-Level Control Scheme: Phase-based Human–Machine Sensors: FSR, Encoder, IMU, EMG Low-Level Control Scheme: Adaptive (active disturbance rejection) Actuation Mechanism: Brushless DC Motors
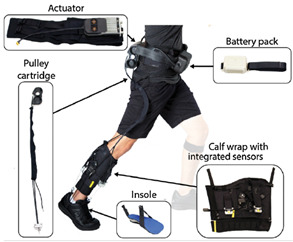 Sloot et al., 2018 [41] (copyrights authorized by Elsevier)	Purpose: General Augmentation Bilateral: Yes DoF: 1 DoF Plantar Flexion Portability: Tethered Weight: 3.8 kg	High-Level Control Scheme: Phase-based Human–Machine Sensors: Angle Sensor, IMU Low-Level Control Scheme: Simple position control Machine–Machine Sensors: Load Cell, Actuation Mechanism: Brushless DC Motor
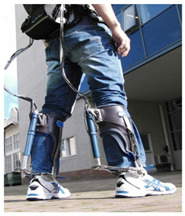 Emmens et al., 2018 [42]	Purpose: Assistive Device: Patients with Spinal Cord Injuries Bilateral: Yes DoF: 1 DoF Dorsiflexion/Plantar Flexion Portability: Portable Weight: 6.7 kg	High-Level Control Scheme: Reflex Model-based Human–Machine Sensors: FSR, Encoder, EMG Low-Level Control Scheme: P, PI Machine–Machine Sensors: Encoder Actuation Mechanism: Brushless DC Motor
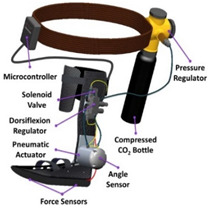 Boes et al., 2018 [43] (copyrights authorized by Elsevier)	Purpose: Assistive Device: Multiple Sclerosis Patients Bilateral: No DoF: 1 DoF Dorsiflexion/Plantar Flexion Portability: Portable Weight: 3.1 kg	High-Level Control Scheme: Phase-based Human–Machine Sensors: FSR, Encoder Low-Level Control Scheme: Proportional Pressure Regulators with Solenoid Valves Machine–Machine Sensors: Pressure Sensors Actuation Mechanism: Pneumatic Cylinders

**Table 2 sensors-22-02244-t002:** High-level control strategies used in PAEs.

High-Level Control Scheme	Reference
Phase-based	[10,37,38,39,40,41,43,49,53,54,55,56,57,58,59,60,61,62,63,64,65,66,67,68,69,70,71,72,73,74,75,76,77,78,79,80,81,82,83,84,85,86,87,88,89,90,91,92,93,94,95,96,97,98,99,100,101,102,103,104,105,106,107,108,109,110,111,112,113,114,115,116,117,118,119,120,121,122,123,124,125,126,127,128,129,130,131,132,133,134,135,136,137,138,139,140,141,142,143,144,145,146,147,148,149,150,151,152,153,154,155,156,157,158,159,160,161,162,163,164,165,166,167,168,169,170,171,172,173,174,175,176,177,178,179]
Impedance-based	[74,75,102,136,161,162,163,164,165,180]
Metabolic-rate-based	[70,96,132,181,182]
Reflex-model-based	[68,183,184,185,186,187,188]
Proportional-myoelectric-based	[11,49,68,89,189,190,191,192,193,194,195,196,197,198,199,200,201]
Adaptive gain proportional-myoelectric-based	[133,134,135,202,203,204,205]
Myoelectric neuromechanical-model-based	[206,207]
Push-button	[49,50]

**Table 3 sensors-22-02244-t003:** Parameters measured from the human user by human–machine interface sensors as reported by the reviewed PAEs. Different sensor types employed for measuring each parameter and their corresponding references are provided in the second and third columns, respectively.

Measured Parameter	Sensor	References
Gait events	FSR	[37,38,39,40,43,56,57,59,60,61,62,63,64,65,67,76,77,78,79,80,81,82,83,84,85,87,94,107,116,117,118,119,120,141,142,143,145,147,148,149,150,151,152,153,156,159,160,166,167,168,170,171,173,174,175,177,178,184,186,187,189]
	Footswitch	[10,49,58,69,72,73,88,89,91,92,93,97,98,99,111,113,122,139,154,155,163,164,165,181,198,199,202]
	IMU	[57,66,74,75,102,158,170]
	Gyroscope	[112]
	Piezoresistive sensor	[109,110]
Ankle joint angle	Encoder	[38,40,42,58,62,63,64,68,69,72,73,85,86,100,103,104,106,108,109,110,117,118,138,139,142,149,150,151,159,160,161,162,165,173,178,181,185,186,187,188,202,206,207,210],
	Potentiometer	[10,43,59,101,102,114,143,147,148,154,155,166,167,170,171]
	Gyroscope	[172]
	Linear displacement sensor	[97,98,99]
	IMU	[64,127,152,211]
	Goniometer	[180,183]
	Attitude sensor	[116]
	Custom strain sensor combined with IMU	[174,175]
	Strain sensor	[87]
Knee joint angle	IMU	[206,207]
	Strain sensor	[87]
Absolute shank angle	IMU	[40,85,87,117,118,159,160,177]
Orientation of shank, thigh, and trunk	IMU	[42]
	Inclinometer	[82]
Angular velocity	Gyroscope	[123,124,125,126,128,129,150,151,157,172]
	IMU	[114,119,120]
Translational acceleration of wearer	IMU	[40,117,118,119,120,159,160]
Foot tilting	Accelerometer	[141]
	IMU	[119,120]
Walking speed	Speed encoder	[204]
Ground reaction force	Force sensor	[10,136]
	FSR	[101,117,131]
Muscle activity	EMG	[11,40,49,68,89,116,117,133,134,135,181,185,186,190,191,192,193,194,196,197,198,199,200,201,202,203,204,205,206,207]
Anatomical ankle generated torque	Strain gauges	[101,102]
Exoskeleton frame–user interaction forces	Force sensor	[136]
Respirometry	Metabolic mask	[70,96,132]

**Table 4 sensors-22-02244-t004:** Detailed technical information of different human–machine interface sensors used in the reviewed PAEs including the specific sensor type, measured parameter, and sensor placement location.

Sensor	Specific Sensor Details	Measurement	Location	Reference
IMU	IMU (gyroscope and accelerometer)	Ankle joint angle	Foot and calf	[56]
	WT901C485, WitMotion, Shenzhen, China	Gait cycle	Shoe	[57]
	EBIMU-9DoFV5, E2BOX Inc., Shanghai, China	Ankle joint angle	Shin and thigh parts of the exoskeleton	[64]
	6-DoF IMU, 100Hz	Gait phase	Shank	[66]
	BNO055 (Bosch, Germany)	Gait phase	Foot	[74,75]
	3DM-GX4-25-RS232-SK1, LORD MicroStrain, Inc., Williston, VT, USA	Absolute shank angle	Main structure	[85]
	MW-AHRS, NTRexLAB	Absolute shank angle	Shank	[87]
	EBIMU-9DoFV4, E2BOX	Shank angle in sagittal plane	Medial shank	[177]
	3 × Xsens (Xsens Technologies B.V., Enschede, The Netherlands)	Orientation of the shank, thigh, and trunk	Shank, thigh, and trunk	[42]
	MPU6050	Ankle joint angle	Foot	[211]
	Not specified	Gait phase segmentation	Foot	[102]
	IMU (Shimmer Inc., Dublin, Ireland )	Angular velocity	Shank	[114]
	SN-IMU5D-LC, Cytron, Simpang Ampat, Malaysia	Shank’s angular velocity in the sagittal plane and accelerations along the y and z axes.	Mechanical structure, near shank	[158]
	2 × Xsens (Xsens Technologies B.V., Enschede, The Netherlands)	#1: Angle between the shank and the vertical axis #2: Translational acceleration of the wearer along the three axes.	#1 Shank #2 Foot	[40,117,118,159,160]
	Mpu6050 6-axis MotionTracking^TM^ device, InvenSense, San Jose, CA, USA	Leg linear acceleration	Leg brace	[119,120]
	MTi-3, (Xsens Technologies B.V., Enschede, The Netherlands)	Foot angle and angular velocity	Lateral side of the shoe	[127]
	Link, Xsens, The Netherlands	Knee joint angle	Not specified	[206,207]
	XSens MTi-28A53G35, (XSens Technologies. Enschede, The Netherlands)	Orientation and position of the exoskeleton	Medial side of the exoskeleton	[170]
	SEN-09623, 9DoF Razor IMU, Sparkfun Electronics, Boulder, CO 80301, USA.	Orientation of lower leg and foot	Foot and lower leg	[174,175]
	IMU (Sparkfun Electronics, Boulder, CO 80301, USA, with a gyroscope ADXRS610 and two accelerometers ADXL320, from Analog Devices)	Absolute position of the exoskeleton	Not specified	[152]
Gyroscope	Gyroscope	Shank angular velocity to identify heel contact	Not specified	[157]
	Single axis Gyroscope	Gait phase	On the shin	[112]
	2 × Single axis Gyroscope (LY3100ALH, STMicroelectronics, Geneva, Switzerland)	Sagittal angular velocity of the shank and foot	One at the top of the mid-foot and the other at the anterior side of the shank	[123,125,129]
	LY3100ALH, STMicroelectronics-single axis	Angular motion of the foot for gait segmentation	Top of the mid-foot	[124]
	Sparkfun, NIWOT, CO, USA	Angular motion of the foot for gait segmentation	Integrated in the shoe	[126,128]
	2 × Gyroscopes	Sagittal angular motion and velocity of the foot for gait segmentation	Not specified	[172]
	Rate gyro	Angular velocity of shank	Not specified	[150,151]
Accelerometer	Tilt sensor	Tilt of foot	Not specified	[141]
Attitude sensor	2 × JY901 attitude sensors	Ankle joint angle	Parallel to the lever and shank	[116]
	angular	Ankle joint angle and angular velocity	Ankle joint	[113]
Foot pressure sensors	3 × Membrane pressure sensors	Plantar pressure distribution for gait cycle detection	Integrated insole	[56]
	Insole-shaped foot pressure sensors (RX-ES39, Roxi Technology, Jiangsu, China)	Identify the gait state using pressure of three parts, i.e., forefoot, toe, and heel	Shoes	[57]
	4 × FSR (MA-152, Motion Lab System Inc., Baton Rouge, LA, USA)	Ground contact, gait phase	Heel, hallux, first metatarsal head, and fifth metatarsal base	[59,60,61]
	FSR sensor	Gait phase	Heel and big toe	[38,62,63]
	3 × FSR sensors	Gait phase	Toe, heel, and medial of the insole	[64]
	2 × FSR (FlexiForce A401, Tekscan, Boston, MA, USA)	Gait phase	Heel and the metatarsal bone	[65,184]
	FSR sensor	Gait cycle	Under the arch support of the shoe	[67]
	2 × FSR sensor	Gait phase	Under the ball and heel of the foot	[76,77,78,79,81,82,84]
	2 × FSR (FlexiForce A201, Tekscan, Inc., Boston, MA, USA)	Ground reaction force	Under forefoot	[37,80,83]
	2 × FSR (FlexiForce A301, Tekscan, Inc., South Boston, MA, USA)	Gait phase	Embedded into the insole	[85,189]
	Toe contact sensor like pressure switch or force-sensing resistor	Gait timing	Not specified	[94]
	3 × FSR (FlexiForce, Tekscan, Boston, MA, USA)	Ground contact of each foot	Insole	[87]
	Custom-designed FSR sole	Gait phase	Beneath the foot brace	[177]
	FSR (Interlink 406, Adafruit, New York, NY, USA)	Gait phase	The user’s shoe at the anterior and posterior ends of the shoe insoles	[156]
	FSR-151AS pressure sensor (IEE, Contern, Luxembourg)	Heel strike	Heel	[178,186,187]
	2 × FSR sensors	Ground reaction force	Heel and toe	[101]
	FSR (SEN-09376 Antratek used with Phidgets Voltage Divider 1121)	Initiation of new step	Heel	[107]
	IMS009-C7.5 (FSR)	Heel strike	Heel	[116]
	3 × FSR in a force sensitive resistor matrix (FSRM)—(Tekscan, Inc., Boston, MA, USA)	Distribution of ground reaction force	Heel, hallux, fifth metatarsal phalange joints	[40,117,118,159,160]
	FSR	Heel strike	Not specified	[39]
	2 × FSR-402 (Interlink Electronics Inc., Camarillo, CA, USA)	Foot loading pattern as an on/off switch.	Forefoot and heel	[119,120]
	4 × FSR-402 (Interlink Electronics Inc., Camarillo, CA, USA)	Ground reaction force	Below the shoe insoles at the heels and toes	[131]
	2 × FSR-402, 0.5 in circle; (Interlink Electronics Inc., Camarillo, CA, USA)	Gait phase	Heel and metatarsal heads	[43,166,167,168,170,171]
	2 × FSR	Gait phase	Toe and heel	[141]
	4 × FSR (FlexiForce-A201-25lb, Tekscan Inc., Boston, MA 02127, USA)	Gait phase	Embedded in a shoe insole	[174,175]
	Sparkfun SEN-09375)	Gait phase	Heel of the plate	[142]
	FSR	Heel and toe contact	In the shoes	[143]
	3 × FSR (FlexiForce-A201-25lb, Tekscan Inc., Boston, MA 02127, USA)	Ground reaction force	Heel, lower forefoot, and big toe	[145]
	FSR	Heel strike	Heel	[146,147,148,149,150,151]
	3 × FSR	Gait phase	Under heel, middle, and front part of the shoe	[152]
	4 × FSR	Gait phase	Flexible insole	[153]
Force sensor	Ultraflex system—with 6 capacitive force transducers 25 mm square and 3 mm thick	Ground reaction force	Bottom of the exoskeleton, two sensors beneath the heel and four beneath the forefoot region.	[10]
	2 × Force sensors	Interaction forces—the ground reaction forces during the contact of robotic device with the ground and other force sensors measure the interaction forces between the shank of the user and the robotic device.	Not specified	[136]
	GRF sensing system consisting of two force sensors	Gait phase	Integrated into shoe	[173]
Footswitch	McMaster-Carr, Aurora, OH, USA	Heel strike	In the heel of the shoe	[58,69,72,73,181,202]
	Footswitch	Foot contact	Not specified	[49]
	Footswitch	Foot contact	Under left forefoot inside the shoe	[88]
	Footswitch (B&L Engineering, Santa Ana, CA, USA)	Foot contact	Inside shoe	[89,199]
	IP67, Herga Electric, Suffolk, UK	Foot contact	Heel	[91,92]
	Multimec 5E/5G, Mec, Ballerup, Denmark	Foot contact	In the heel of the shoe	[93,97,98,99]
	Footswitch, model MA-153	Heel strike	In the heel of a shoe worn with the orthosis	[10]
	FSW (B&L Engineering, Santa Ana, CA, USA)	Heel and toe contact	Not specified	[111,113]
	B&L Engineering, Santa Ana, CA, USA	Heel strike	Under foot	[122]
	Footswitch (B&L Engineering, Santa Ana, CA, USA) with 4 individual footswitches	Gait phase	Inside shoe—at the heel, forefoot, medial, and lateral zones at the level of metatarsals.	[163,164,165]
	Pressure sensor (footswitch)	Heel strike moment and stride length	Under the shoe	[139]
	2 × Tactile Arrays	Position of orthosis	Incorporated in the foot part of the exoskeleton and in the insole of the healthy leg	[154,155]
Potentiometer	Rotary potentiometer	Ankle joint angle	Attached to the hinged ankle joint of the exoskeleton	[59,60,61]
	Precision potentiometer (resolution of 0.5°)	Ankle joint angle	Exoskeleton ankle joint	[101,102]
	Bourns 6637S-1-502 5-k rotary potentiometer	Ankle joint angle	Not specified	[10]
	Motorized linear potentiometer	Ankle joint angle	Integrated in wearable ankle robot	[114]
	Rotary potentiometer 53 Series, Honeywell, Golden Valley, CA, USA).	Ankle joint angle	Exoskeleton ankle joint	[43,166,167,170,171]
	Linear potentiometer	Ankle joint angle	Exoskeleton ankle joint	[143,148]
	linear and an angular potentiometer	Ankle motion	Not specified	[147]
	Rotary potentiometer	Ankle joint angle	Exoskeleton ankle joint	[154,155]
Encoder	Optical encoder (E8P; US Digital, Vancouver, WA, USA)	Ankle joint angle	Exoskeleton ankle joint	[58]
	2 × Absolute encoders (AMT203-V, CUI Inc., Tualatin, OR, USA)	Ankle joint angle	Exoskeleton joints corresponding to the talocrural and subtalar joints	[38,62,63,64]
	Digital optical encoder	Ankle joint angle	Exoskeleton ankle joint	[68]
	E4P and E5(US Digital Corp., Vancouver, WA, USA), for alpha and beta exoskeleton	Ankle joint angle	Exoskeleton ankle joint	[69]
	Absolute magnetic encoder (MAE3, US Digital, Vancouver, WA, USA)	Ankle joint angle	Lateral side of each exoskeleton’s ankle joint	[73,202]
	Digital optical encoders (E5, US Digital, Vancouver, WA, USA)	Ankle joint angle	Exoskeleton joint shaft	[72,181]
	Angular sensor, PandAuto P3022, Mexico, Mexico	Absolute angle of Link 1 in exoskeleton	Exoskeleton	[86]
	Optical incremental encoder (2048 CPR, E6-2048-250-IE-S-H-D-3, US Digital, Inc.)	Ankle joint angle	Exoskeleton ankle joint	[85]
	RMB20IC13BC SSI-encoder (RLS-Renishaw, Ljubljana, Slovenia)	Ankle joint angle	Exoskeleton ankle joint	[42,100,178,185,186,187]
	Incremental optical encoder (US Digital HUBDISK-2-2000-625-IE, module EM1-2-2000-I, DI/O type, 5 pins, 5V)	Ankle joint angle	Exoskeleton ankle joint	[103,104,106,210]
	Joint encoder (2000 CPT, HEDS-5600, Broadcom, San Jose, CA, USA), quadrature encoder-70	Ankle joint angle	Lateral 3D-printed mount on exoskeleton ankle joint	[108,109,110]
	Encoder E2	Ankle joint angle	Exoskeleton ankle joint	[188]
	Incremental encoder	Ankle joint angle	Exoskeleton ankle joint	[40,117,118,159,160]
	linear incremental encoders (Renishaw, Chicago, IL, USA)	Ankle joint angle	Traction drive	[161,162,165]
	Optical 3 phase 4000 CPR	Ankle joint angle	Exoskeleton ankle joint	[131]
	Absolute rotary encoder 20 b Aksim, RLS (Renishaw), Kemnda, Slovenia).	Ankle joint angle	Exoskeleton ankle joint	[206,207]
	Optical encoder (US Digital Inc.)	Ankle joint angle	Exoskeleton ankle joint	[138,139]
	Magnetic encoder (AN25-analog, KD Mechatech Co., Korea)	Ankle joint angle	Exoskeleton ankle joint	[173]
	Rotary encoder	Foot rotation	Base of shank	[142]
	Absolute angular encoder	Ankle joint angle	Exoskeleton ankle joint	[149,150,151]
Goniometer	Goniometer (5 kHz, 250 Hz Biometrics, Newport, UK)	Ankle joint angle	Exoskeleton ankle joint	[180]
	Goniometers (500 Hz, Biometrics, Newport, UK)	Ankle joint angle	Exoskeleton ankle joint	[183]
Linear displacement sensor	100 Hz; SLS130, Penny & Giles, Christchurch, Newport, UK	Ankle joint angle	Foot and shank sections of the exoskeletons	[97,98,99]
Strain sensor	Soft strain sensor	Ankle and knee joint angle	Knee and the ankle joints	[87]
	4 × strain gauges connected to a full Wheatstone bridge	Human–exoskeleton interaction torque	On the exoskeleton frame, near the ankle joint	[101,102]
	4 × custom-built strain sensors	Ankle joint angle	Dorsal and medial side of the ankle	[174,175]
Piezoresistive sensor	3 × Piezoresistive sensors	Gait phase	Foot section of the exoskeleton, underneath the calcaneus, the first metatarsal head, and the hallux.	[109]
	Piezoresistive sensor	Heel strike	Underneath the calcaneus	[110]
EMG	Surface electrodes, high-pass filtered at 20 Hz, rectified, low-pass filtered at 6 Hz	Muscle activity	Gastrocnemius muscle	[68]
	2 × Wired, bipolar electrodes (Bagnoli Desktop System, Delsys Inc., Boston, MA, USA)	Muscle activity	Medial and lateral aspects of the soleus	[202]
	Wireless EMG system (Bagnoli, Delsys, MA, USA)	Muscle activity of four lower-leg muscles on the exoskeleton side	Medial gastrocnemius, lateral gastrocnemius, soleus, and tibialis anterior	[181]
	1200 Hz, TeleMyo, Noraxon USA, Scottsdale, AZ, USA	Muscle activity	Soleus and tibialis anterior	[11]
	1200 Hz, Konigsberg Instruments, Inc., Pasadena, CA, USA	Muscle activity	Soleus, medial gastrocnemius, tibialis anterior	[49,89,190,191,192,193,194,196,197,198,199,200,203]
	Surface EMG (960 Hz SX230, Biometrics, Newport, UK).	Muscle activity of the paretic side	Soleus	[201,204]
	Surface EMG	Muscle activity	Tibialis anterior and soleus muscles	[185,186]
	Not specified	Muscle activity	Tibialis anterior, gastrocnemius, soleus, and the rectus femoris	[116]
	2 × Electromyography sensors (Delsys)	Muscle activity	Tibialis anterior and gastrocnemius	[117]
	EMG surface electrodes-1000 Hz; SX230, Biometrics	Muscle activity	Soleus	[133,134,135]
	AxonMaster 13E500, Ottobock, Germany	Muscle activity	Lower limb	[206,207]
	Surface EMG	Muscle activity	Tibialis anterior, lateral gastrocnemius, medial gastrocnemius, peroneus longus, and soleus	[205]

**Table 6 sensors-22-02244-t006:** Low-level control strategies used in PAEs.

Low-Level Control Scheme		References
Classical PID		PID [39,53,54,55,56,67,68,76,77,78,79,82,86,101,102,108,109,110,114,119,120,121,136,138,139,140,154,155]
		P [100,106,107,113,146,172,178,185,186,187,204]
		PI [85,100,105,178,185,186,187]
		PD [10,57,65,66,69,70,71,72,73,81,137,147,148,149,150,151,152,153,161,162,163,164,165,180,181,202,206,207]
		Adaptive PID [37,64,80,83,84]
Iterative Learning		[58,68,69,70,71,72,73,131,181,182,202]
Adaptive		[40,68,159,160,189]
Sliding Mode		[53]
Open-Loop Feed-Forward		[94,158,184,206,207]
Pneumatic Actuation Controls	On-Off Solenoid Valves	[146,156,166,167,170,171,173,174,175]
	Pulse Width Modulation (PWM) with Solenoid Valves	[38,62,63,141,173,188]
	Proportional Pressure Regulators with Solenoid Valves	[11,43,49,50,88,89,90,91,92,93,95,96,97,98,99,132,133,134,135,142,168,169,190,191,192,193,194,195,196,197,198,199,200,201,203]

**Table 7 sensors-22-02244-t007:** Parameters measured by machine–machine interface sensors deployed in PAEs as reported by the reviewed articles. Different sensor types employed for measuring each parameter and their corresponding references are provided in the second column.

Measured Parameter	Sensor
Cable/rope tension	Tension sensor [53,182] force sensor [57] load cell [39,58,69,108,109,110,122,123,124,125,126,127,128,129,180,183,204] strain gauge [72,113,181]
Mechanical deflection	Potentiometer [10,55,67,108,110,137,146,153,176] encoder [206,207,236]
Pressure	Pressure sensor [62,63,146,156,158,166,171,173,176,232]
Pneumatic muscle force	Load cell [11,38,49,50,62,63,88,89,91,96,97,98,99,131,134,190,191,192,193,194,196,197,198,199,200,203]
Real torque measured from actuator	Torque sensor [85]
Reaction torque measured at the ankle	Torque sensor [37,76,77,78,79,80,81,82,83,84,104,105,106] linear potentiometer [234] strain gauges [68,69,73,202]
Forces delivered by exoskeleton	Load cell [138,139]
Motor current	Current sensor [39,85,161,162]
Motor position/velocity	Encoder [39,55,57,59,67,68,72,108,110,111,112,113,115,116,121,141,143,147,148,149,150,151,161,162,165,172,180,181,188,206,207,235] Hall sensor and resolver [85]
Cable position	Potentiometer [126]
Motor stroke	Encoder [42,100,178,185,186,187]
Slave cylinder stroke	Optical distance sensor [158]
Actuator lever/link angles	Encoder [103,104,106,210]

**Table 8 sensors-22-02244-t008:** Detailed technical information of different machine–machine interface sensors used in the reviewed PAEs, including the specific sensor type, measured parameter, and sensor placement location.

Sensor	Specific Sensor Details	Measurement	Location	Reference
Load cell	LC201 Series; OMEGA Engineering, Stamford, CT, USA	Cable tension	At the ankle	[58]
	Inline tensile load cell: DCE-2500N, LCM Systems, Newport, UK,250Hz LP Filter	Actuation force	Attached to the end effector moment arm (~ 10 cm) through a series elastic element	[180]
	Load cells (500 Hz, LCM Systems Ltd., Newport, UK)	Cable tension	In series with the force transmission cables and series elastic element	[183]
	Tension load cell (CDFS-200, BONGSHIN LOADCELL, Osan, Korea)—one with each pneumatic artificial muscle	Pneumatic muscle force	Attached at the end of the pneumatic muscle	[38,62,63]
	LC201, OMEGA Engineering Inc., Stamford, CT, USA)	Bowden cable tension	Alpha exoskeleton, Bowden cable	[69]
	Tension load cell-(LC8150-375-1K 0–100 lbs, 1200Hz, OMEGA Engineering, Stamford, CT, USA)	Pneumatic muscle force	Between the pneumatic muscle and the rod end	[11,49,50,88,89,96,190,191,192,193,194,196,197,198,199,200,203]
	Load cell (W2, A.L. Design, Buffalo, NY, USA)	Tensile force of pneumatic muscle	Not specified	[91]
	100 Hz; 210 Series, Richmond Industries Ltd., Reading, UK	Pneumatic muscle force	Connected between the orthoses and the pneumatic muscles	[97,98,99]
	Two in-line load cells (LSB200, FUTEK Advanced Sensor Technology, Irving, CA, USA)	Actuation force	Posterior side of the calf, near the proximal part of the orthosis.	[108,109,110]
	Inline tensile load cell (DCE-2500N, LCM Systems, Newport, UK)	Actuation force	Bowden cable	[204]
	LFT-13B, Shenzhen Ligent Sensor Tech Co., Ltd., Shenzhen, China (inline load cell)	Force applied on the struts	Not specified	[39]
	2 × Load cell—one cantilevered load cell (Phidgets 3135 50 kg Micro Load Cell), second load cell (LCM300, FUTEK Advanced Sensor Technology, Irving, CA, USA)	Cable force at the top of the Bowden cable and forces delivered to the wearer	First one in pulley module, second one at the ankle in series with the cable	[122]
	LTH300, FUTEK Advanced Sensor Technology, Irving, CA, USA	Bowden cable force	In series with the Bowden cable and the calf wrap	[123,124,125,129]
	LSB200, FUTEK Advanced Sensor Technology Irving, CA, USA	Assistive force transmitted to the hip joint via straps	Left side of exosuit in series with the two vertical straps and the waist belt	[124,129]
	2 × LSB200, FUTEK Advanced Sensor Technology, Irving, CA, USA	Delivered force at the ankle—DF and PF forces generated by Bowden cable retractions	Integrated into the exosuit’s textile loops of the calf wrap	[126,127,128]
	Not specified	Pneumatic muscle force	Between the NcPAM and the bottom plate	[131]
	OMEGA Engineering, Stamford, Connecticut	Actuation kinetics	In series with actuator	[134]
	Load cell (range +/− 220 N; Transducer Techniques Inc.)	Forces transmitted to wearer	At the extremity of the slave cylinder	[138,139]
Strain gauge	Not specified	Plantar flexion torque	On heel lever	[68,73]
	4 ×strain gauges (MMF003129, Micro Measurements, Wendell, NC, USA) in a Wheatstone-bridge	Torque	On ankle lever	[69,202]
	Wheatstone bridge consisting of four strain gauges (KFH-6-350-C1-11L1M2R, OMEGA Engineering, Norwalk, CT, USA)	Assistive torque	End of titanium ankle lever	[72,181]
	LCM200, FUTEK Advanced Sensor Technology, Inc., Irvine, CA, USA	Cable tension	On transmission cable	[113]
Encoder	Incremental encoder	Motor position and velocity	Motor	[55]
	AMT103-V, CUI Inc., OR, USA (2048 counts per revolution).	Motor position and velocity	Motor	[57]
	Optical encoder (E5 Optical Encoder, US Digital, Vancouver, WA, USA)	Motor pulley velocity	Motor	[180]
	Not specified	Motor position and velocity	Motor	[59,60,61]
	Incremental encoder with 1024 count per turn	Motor position	Motor	[67]
	Digital optical encoders	Motor position	Motor	[68]
	Digital optical encoders (E5, US Digital, Vancouver, WA, USA)	Motor position	Motor shaft	[72,181]
	Incremental encoder (SCH24-200-D-03-64-3-B, Scancon, Allerød, Danmark)	Motor stroke	Motor axis	[42,100,178,185,186,187]
	Absolute magnetic encoder (AS5048A, SPI type, 6pins, 5V, a4-bit)	Torque angle/lever arm angle	In actuator	[103,104,106,210]
	Motor-shaft encoder	Motor position and velocity	Motor	[108,110]
	Encoder E1	Motor position	Motor	[188]
	500-Count quadrature incremental optical encoders (model: HEDL 5540, Maxon Motors, Sachseln, CH).	Motor position	Motor	[111,112]
	14-Bit magnetic on-axis relative encoder (AS5047P and AS5047D, AMS AG, Premstaetten, AT)	Motor position	Motor	[113]
	Encoder (ENX16 EASY 500IMP)	Motor position	In actuator module	[115,116]
	Motor Encoder	Motor position and velocity	Motor	[39,143,147,148]
	Quadrature encoders (2×195 RPM and 2×60 RPM HD premium planetary gearheads)	Motor position and velocity	Motor	[121]
	Rotary encoder—Gurley R119 rotary encoders (Gurley, Troy, NY)	Commutate the motor	Mounted coaxially with the motors	[161,162,165]
	Encoder FPC optical 3 phase 4000 CPR	Not specified	Inside the thrust bearing	[131]
	Absolute angle Hall encoder (MHM, IC Haus, Germany)	Motor position	Motor	[206,207]
	Angle encoder	Motor position and velocity	Motor	[235]
	Absolute rotary encoder 20 b AksIM, RLS (Renishaw), Kemnda, Slovenia).	Spring deflection	Not specified	[206,207]
	Encoder (5540 HEDL)	Motor position	Motor	[172]
	2 × Rotary encoder and linear encoder	Spring deflection	In actuator	[236]
	Encoder with servo motor	Actual position sensing of actuator	Motor	[141]
	Digital incremental motor encoder	Determine position of lead screw	Not Specified	[149,150,151]
Hall sensors and resolver	Not specified	Motor position	Motor	[85]
Potentiometer	Linear potentiometer	Spring deflection	Motor housing	[55]
	Linear potentiometer (50 mm travel length)	Spring deflection	Assembled to stainless steel pipe with a 3D-printed plastic housing.	[67]
	Linear potentiometer	Spring deflection	Top of spring module	[10,108,110]
	Linear potentiometer (P3 America Inc., San Diego, CA, USA)	Cable position	With actuator	[126]
	Linear potentiometer	Joint torque	Mounted in parallel with series spring	[234]
	Softpot linear position sensor	Transpose of spring	Not specified	[137]
	Linear potentiometer	Deflection of links	Upper part of actuator link	[146,176]
	Linear sliding potentiometer	Spring deflection	Fixed in the two-support platforms of the springs.	[153]
Tension sensor	Not specified	Cable tension	Bowden cable	[53]
	Not specified	Cable tension	At the end of Bowden cable	[182]
Pressure sensor	TST-20.0, TIVAL Sensors GmbH, Wuppertal, Germany	Pressure of pneumatic muscle	Pneumatic muscle	[62,63]
	ASDXAVX 100PGAA5, Honeywell Sensing and Productivity Solutions, Charlotte, NC, USA	Actuator pressure	Actuator	[156,232]
	Pressure sensors (PX3AN1BH667PSAAX, Honeywell)	Measure the pressure in each hydraulic transmission	Not specified	[158]
	Tethered pressure transducer: 4100 series, American Sensor Technology; Mt. Olive, NJ, USA)	Assistive torque-pressure in actuator	In actuator chamber	[166]
	Pressure transducers (AST4000A00150P3B1000, 150 psig and AST4000A00100P3B0000, 100 psig, American Sensor Technologies, Inc, Mount Olive, NJ, USA)	Compressed CO_2_ pressure on both sides of the actuator	Actuator	[171]
	5V G1/4 0–1.2 MPa, China	Senses the pressure in the cylinder chamber system	With the control hardware, attached to the waist of user	[173]
	Not specified	Not specified	Upper end of actuator	[146,176]
Torque sensor	TRT-500, Transducer Techniques, Temecula, CA, USA	Reaction torque provided by the motors through the ankle pulley	Placed in line with each exoskeleton ankle joint/mounted on the insole	[37,76,77,78,79,80,81,82,83,84]
	Torque sensor (TPM 004+, Wittenstein, Inc., Igersheim, Germany)	Actuator torque output	Installed between the actuator case and the main structure	[85]
	DRBK, ETH Messtechnik, 200 Nm, 0.0122 Nm resolution	Actuator torque output	Attached to test setup	[104,105,106]
Current sensor	Not specified	Active current	Motor	[85]
	Not specified	Motor current	Motor	[39]
	Analog current sensor (Interactive Motion Technologies board employing TI/Burr-Brown 1NA117P)	Motor current to estimate motor torque	Motor	[161,162]
Distance sensor	Optical sensor (GP2Y0A51SK0F, Sharp)	Slave cylinder stroke	Not specified	[158]
Force sensor	ZZ210-013, Zhizhan Measurement and Control, Shanghai, China	Cable force	Heel cable and forefoot cable	[57]

## Data Availability

Not applicable.

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
