# Peer review of "Application of Wearable Sensors in Actuation and Control of Powered Ankle Exoskeletons: A Comprehensive Review"

_sensors, 2022, doi:10.3390/s22062244_

Round 1

Reviewer 1 Report

This paper reviews the control scheme and driving principle of PAEs, and the contribution of wearable sensors to the realization of PAEs. The paper can be considered for acceptance if the authors can address the following points:

  1. The statements of “The 14 articles published between January 2000 and June 2021 by PubMed, SAGE journals, IEEE Xplore, 15 Scopus, ScienceDirect, Web of Science were searched for the following keywords: [‘ankle’+ ‘exoskel-16 eton’] or [‘ankle’+ ‘robot’] or [‘ankle’ + ‘power’+ ‘orthosis’]. After removing duplicates, review pa-17 pers and multi-joint exoskeletons, the remaining articles were reviewed” is not necessary (actually rarely seen) to be included in the abstract.
  2. It seems not a common practice to include the review method illustration and relevant flow chart in a review paper.
  3. For the first reviewed work in table 1, the authors mentioned “Machine-Machine Sensors: Torque Sensors”. However, we can see from the physical frame diagram that the Machine-Machine Sensors section includes not only Torque Sensors, but also Force Sensors.
  4. The signal processing flow chart in Page 12 is incomplete. The output form of sensor is mostly current output and voltage output form, which belongs to analog signal. Analog signal acquisition and ADC should be included. Besides, analog filter also play a critical role.
  5. There are some format issues to be revised. For instance, in Page 12 Line 237, the citation of ref 47 should be revised, and the figure caption for the flow chart is missing.

Reviewer 2 Report

R14-R18: I think the authors should remove from the abstract the sentence "The articles published between January 2000 and June 2021 by PubMed, SAGE journals, IEEE Xplore, Scopus, ScienceDirect, Web of Science were searched for the following keywords: [‘ankle’+ ‘exoskeleton’] or [‘ankle’+ ‘robot’] or [‘ankle’ + ‘power’+ ‘orthosis’]. After removing duplicates, review papers and multi-joint exoskeletons, the remaining articles were reviewed." The abstract should be a short description of the technical aspects discussed in the manuscript, not of the methods used to search articles in databases.
-The reference for figure 1 should be added.
R132: the title of the section should be renamed. ("Scientific methods")
R134-149: I think should be removed.
-What reprezent figure 2? The contribution of this manuscript consists in searching techniques based on algorithms or critical opinion concerning the published articles about 'Wearable Sensors in Actuation and Control of Powered Ankle Exoskeletons'?
-The authors should add more comments about tables 3, 4, 7 and 8.

Round 2

Reviewer 2 Report

I recommend this paper for publication.